# Quantifying economic resilience from input–output susceptibility to improve predictions of economic growth and recovery

Peter Klimek [1,2], Sebastian Poledna[2,3] & Stefan Thurner[1,2,3,4]

Modern macroeconomic theories were unable to foresee the last Great Recession and could neither predict its prolonged duration nor the recovery rate. They are based on supply–demand equilibria that do not exist during recessionary shocks. Here we focus on resilience as a nonequilibrium property of networked production systems and develop a linear response theory for input–output economics. By calibrating the framework to data from 56 industrial sectors in 43 countries between 2000 and 2014, we find that the susceptibility of individual industrial sectors to economic shocks varies greatly across countries, sectors, and time. We show that susceptibility-based growth predictions that take sector- and country-specific recovery into account, outperform—by far—standard econometric models. Our results are analytically rigorous, empirically testable, and flexible enough to address policy-relevant scenarios. We illustrate the latter by estimating the impact of recently imposed tariffs on US imports (steel and aluminum) on specific sectors across European countries.

---

[1] Section for Science of Complex Systems, CeMSIIS, Medical University of Vienna, Spitalgasse 23, 1090 Vienna, Austria. [2] Complexity Science Hub Vienna, Josefstädter Strasse 39, 1080 Vienna, Austria. [3] IIASA, Schlossplatz 1, 2361 Laxenburg, Austria. [4] Santa Fe Institute, 1399 Hyde Park Road, Santa Fe, NM 85701, USA. Correspondence and requests for materials should be addressed to S.T. (email: stefan.thurner@muv.ac.at)

In 2008 several advanced economies were hit by the largest recessionary shock in history[1]. This Great Recession was followed by a puzzlingly slow rate of economic recovery[2]. Economists not only got the likelihood of a crisis of this severity wrong, as Paul Krugman famously noted, but also how fast we would recover from it[3]. Efforts to understand the origin of this blind spot in economic theory and its failure to predict systemic events have fueled the interest in how economic systems absorb shocks and how they recover[4–7]. Our lack of understanding of economic resilience[8] has been explained by a fundamental mismatch between macroeconomic theories and the reality of how markets work, especially in the presence of extreme events[9]. General equilibrium theory holds that economic growth is characterized by a balance of demand and supply which results in prices that signal an overall equilibrium[10–12]. However, the crisis was a story of contagion, interdependence, interaction, networks, and trust[9] that led these equilibrium assumptions ad absurdum. The inappropriate use of equilibrium concepts in economics in the context of extreme events was pointed out some time ago[13]. So far, the only way to address and study economic nonequilibrium are highly stylized statistical models of money exchange[14] and computer simulations[15,16].

In physics, nonequilibrium systems are equally hard to understand and control. Aside from some seminal contributions[17–20], a unified framework for out-of-equilibrium phenomena has yet to be found[21]. However, to understand how systems in equilibrium behave in response to shocks has been successfully addressed within the framework of linear response theory (LRT). According to LRT, an external force, $X(t)$, acting on a system induces a proportional flux, $J(t) = \rho X(t)$. The proportionality is given by transport coefficients or susceptibilities, $\rho$, formally related to the decay of the system's equilibrium autocorrelation functions—the so-called Green–Kubo relations[22,23]. LRT provides the theoretical basis for many linear phenomenological laws that constitute the core of each high school physics curriculum, such as Ohm's law, Newtonian viscosity, or magnetic and electric susceptibilities (see Supplementary Note 1 and Supplementary Table 1).

In the following we give an intuitive account of LRT. Imagine someone hands you a serving tray with an elaborate house of cards on it. Which card will fall first and cause the collapse of the house? You try to answer this question by doing minimal damage: you slightly nudge the tray and observe how the cards respond. If a tiny nudge moves certain cards, you might conclude that those are the first to fall if the tray was pushed harder. The first cards that you observe to move you call most susceptible to the shock (nudge). If you observe no movements of cards whatsoever, you might be tempted to apply a stronger kick; the cards could be glued together. A similar way of reasoning underlies the theory of linear response. In the language of statistical mechanics, the tiny nudge initially applied plays the role of equilibrium fluctuations. These fluctuations may or may not move certain cards as a response—they induce a flux. One then assumes that this response is proportional to the magnitude of the nudge, the proportionality being described by transport coefficients.

How Leontief economies respond to shocks has been studied in a number of works briefly summarized as follows[24–28]. Considering a (variant of a) Leontief IO economy, a shock is specified using varying degrees of external assumptions. It is then studied how the economy relaxes to the old or new equilibrium configuration, unless yet another shock is assumed. Our approach follows a completely different strategy. We consider shocks that drive the economy away from its equilibrium into a nonequilibrium stationary state. This stationary state is different from the original equilibrium state and the equilibrium state implied by the perturbed productivity or technology. Instead, the new state is

characterized by the system trying to achieve a balance between two opposing forces, namely a relaxation to the (unaltered) equilibrium state and the external shock that actively drives the system away from equilibrium. We call an economy in such a state a driven economy. Market participants (sectors) in a driven IO economy incorporate the external shock in their production functions without altering their demand or required input from other sectors. The perturbed output of these sectors then propagates along the IO network to other sectors, thereby driving the economy into a new nonequilibrium stationary state.

The underlying ideas of LRT have been exploited in other contexts, such as the theory of linear time-invariant systems, with applications in signal processing and control theory[29]. In econometrics, impulse response functions describe how external shocks drive macroeconomic variables such as output, consumption, or employment in vector autoregressive (VAR) models[30,31]. Instead of studying the relaxation dynamics of macroeconomic variables within highly stylized VAR models, we focus on structural characteristics of dynamical IO matrices that capture the interactions between economic sectors.

In this work we develop an analytic and empirically testable framework for the nonequilibrium response and recovery of severely disrupted economies. For the first time we formulate a theory of linear response for input–output (IO) economics[32,33]. We will show that the LRT rationale can be used to study the response of Leontief IO economies[33] to large shocks. The resulting framework is applied to IO data from 56 industrial sectors in 43 countries between 2000 and 2014[34] (see Supplementary Note 2). We show that the lack of recovery after the Great Recession can be related to the susceptibility of individual sectors. As in statistical physics, in the economic context LRT serves as a firm analytic link between the microscopic equilibrium fluctuations of a system and its macroscopic out-of-equilibrium response to large shocks.

## Results

**Obtaining economic susceptibilities from input–output data**. Our formalism provides a quantitative and data-driven method to benchmark individual countries and production sectors in terms of their economic susceptibilities to shocks (see Methods). To illustrate the method, we measure country- and sector-level economic susceptibilities respectively by using the world input–output database (WIOD)[34]. We consider data for 56 sectors in 43 countries between 2000 and 2014. For each country, $c$, and year, $t$, we extract demands $D_i(t)$, technical coefficients $A_{ij}(t)$, and outputs $Y_i(t)$, where subscripts refer to sectors. Our aim is to compute the economic susceptibility matrix for a country and year, $\rho_{ij}^c(t)$. Here $t$ denotes the year the data were taken to compute $\rho_{ij}^c(t)$. Based on data from $t$, we model output changes forward in time on a scale denoted by $t' > t$. We numerically integrate the stochastic differential equation for a Leontief economy, Eq. (5), in the absence of an external shock ($X_i(t') = 0$ for all $t' \geq t$ and $i$). Now the time-lagged equilibrium correlation functions between two sectors in Eq. (6) can be computed. The entries in $\rho_{ij}^c(t)$ correspond to the area under the curve of these correlation functions when plotted as a function of the time lag in Eq. (6). Susceptibilities of individual sectors $\rho_i^c(t)$ are the columnwise sums of matrix $\rho_{ij}^c(t)$.

Response curves of individual sectors are obtained by integrating the correlation functions under specific shocks. In Fig. 1 we assume an impulse demand shock of unit size applied at time $t' = t$, $X_i(t') = \delta(t' - t)$ in each sector $i$ (Fig. 1a), leading to different response curves for each sector in the USA in 2014 (Fig. 1b). The shock is applied at $t = 2014$ and results in the same large decrease of output in all sectors immediately after $t$. For $t' > t$,

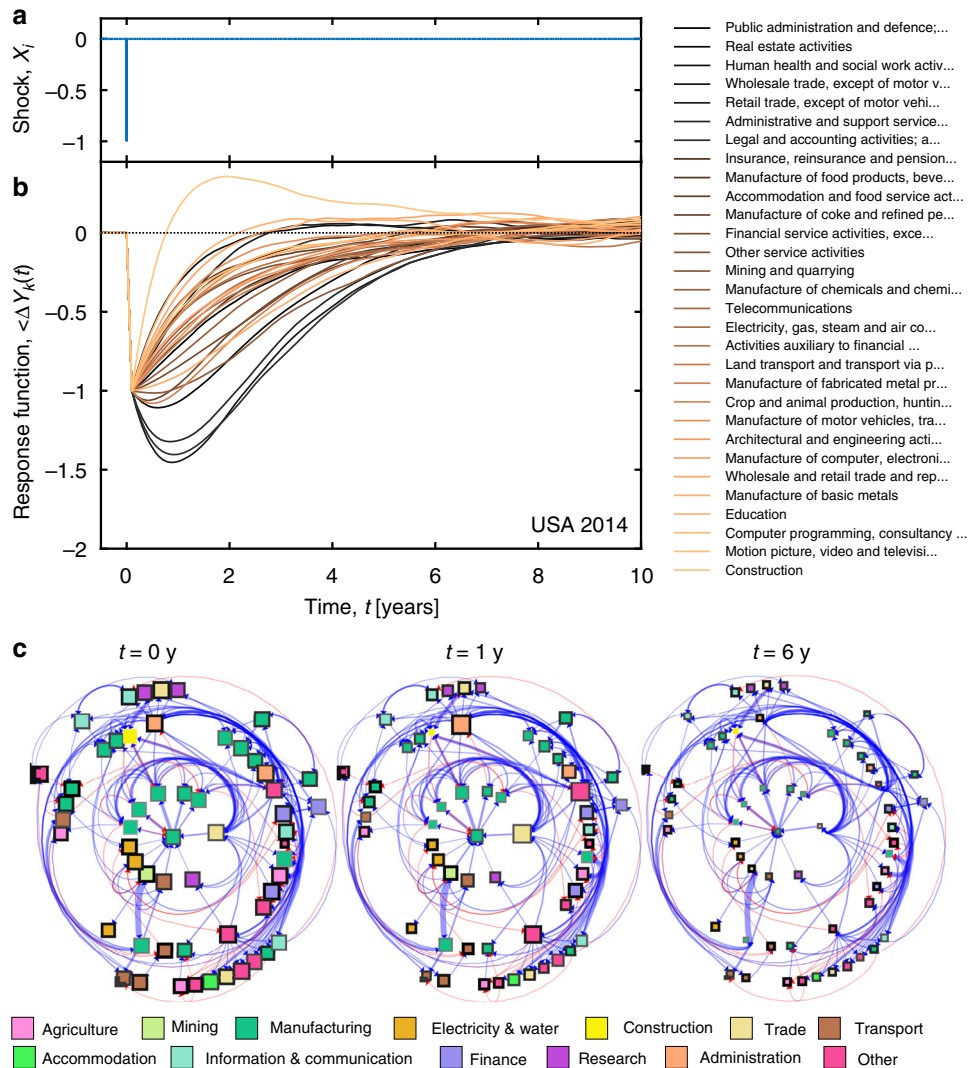

**Fig. 1** Visualization of response curves. **a** An impulse shock of unit size is applied in year $t = 2014$ to every sector, $i$, in the USA. In response, the output of each sector is driven from its equilibrium value, given by $\langle \Delta Y_i(t') \rangle_X = 0$. **b** Every line corresponds to one of the 30 largest sectors, ordered according to their susceptibility to the shock (i.e. the area between the response curve and the dotted line that represents the equilibrium value). The sectors with the largest impact are public administration, real estate activities, human health, and wholesale trade. On the other end of the scale we find the construction sector, that after the initial shock profits from the disruptive event. Note the time scale. Depending on the sector, full economic recovery might take up to 6−10 years. **c** A network visualization of the backbone of the susceptibility matrix $\rho_{ij}^c(t)$ for the USA in 2014 is shown. Nodes are sectors and blue (red) weighted links indicate positive (negative) susceptibilities. Node colors show groups of sectors (see Supplementary Table 2) and thickness of the node border gives the sum of the weights of the incoming links. Node sizes are inversely proportional to the values of the response functions in (**b**) for $t' = t$, $t' = t + 1$ and $t' = t + 6$ years after the shock was applied. Source data are provided as a Source Data file

there appear substantial differences between sectors. For some sectors the shock is amplified, such as for public administration, real estate activities, health, or wholesale trade. Other sectors immediately start to rebound from the shock, for instance, the various manufacturing sectors. The fastest rebound is observed for the construction sector, where production even exceeds the equilibrium level (0) for an extended period of time. Overall, it can take up to 6−10 years for each sector to return to its equilibrium state (sectoral recovery time). Whether a shock is amplified or suppressed in a sector depends on the structure of the susceptibility matrix $\rho_{ij}^c(t)$ (see Fig. 1c). There we show the backbone of $\rho_{ij}^c(t)$ (obtained after applying the disparity filter with $p = 0.05$ [35]) as a directed weighted network. Blue (red) links show positive (negative) susceptibilities. Node colors indicate groups of similar sectors, thickness of the node border is proportional to the sum of the weights of all incoming links (see Supplementary

Table 2); node sizes are inversely proportional to the values of the response functions in Fig. 1b at a particular point in time. We show three snapshots of this network at the time when the initial shock is applied ($t' = t$), and one ($t' = t + 1$), and six ($t' = t + 6$) years afterwards. Figure 1c shows that some but not all of the sectors with a particularly strong shock amplification tend to be among those with a large number of incoming links (and weights thereof), compare for instance the administration (large shock, many incoming links with strong weights) and construction (almost negative shock amplification, small number of incoming links) sectors.

We apply the above procedure for every year $t$ (where the shock is applied), every country $c$, and every sector $i$, to compute a susceptibility value, $\rho_i^c(t)$. The average country susceptibilities, $\rho^c = \langle \rho_i^c(t) \rangle_i$, are obtained by averaging $\rho_i^c(t)$ over all sectors $i$ and years $t$ (see Supplementary Figure 1). The higher the values of $\rho^c$,

the higher is the chance that any sector $i$ in $c$ will be impacted by a shock in any other sector $j$. We find similar levels of susceptibility in a large number of countries across Europe, North America, and China. Substantially smaller susceptibilities are found for Croatia, Greece, Malta, and Luxembourg. For those countries, our findings suggest a higher production concentration in a smaller number of sectors and consequently a smaller exposure to cascading impacts between different sectors (within the country). At the other end of the spectrum, it is striking to see that four out of the five BRICS countries appear as the most susceptible countries, namely Russia, China, India, and Brazil; data for South Africa are not included in the WIOD due to the lack of available data with sufficient quality[36]. This suggests that the sustained above-average growth of these countries in the last 10−20 years did not go along with the formation of resilient economic production structures.

In Supplementary Figure 2 we show the output-weighted average sector susceptibility, $\rho_i = \langle \rho_i^c(t) \rangle_c$ (see also Supplementary Table 2). Sectors with the highest susceptibilities include wholesale trade, administrative services, electricity, and financial service activities. This means that if a country experiences an economic shock, those sectors are most likely to be affected by shocks in other sectors. In contrast, we find that sectors like scientific research, activities of extraterritorial organizations, manufacture of transport equipment, or air and water transport are relatively immune to cascading events.

**Empirical validation of the linear response formalism.** We now show to what extent the economic susceptibility matrix $\rho_{ij}$ is predictive of the size and direction of sectors' future output changes. First, it can be shown that the average size of sectoral output changes can be predicted (out-of-sample) by means of sector-size-dependent random fluctuations. To evaluate the linear response relation $\langle \Delta Y_k \rangle_X = \rho_{ki} X_i$ it is necessary to specify the shock, $X_i$. A particularly simple assumption is that $X_i$ is itself noise with a magnitude proportional to the output of sector $i$, $X_i = \eta_i Y_i$, where $\eta_i$ has the same expectation value in each sector, $\langle \eta_i \rangle_i = \eta$. The hypothesis is that if $\rho$ indeed captures structural characteristics of economies that relate to their recovery from shocks, one should be able to extract how violently $Y_k$ fluctuates in the future, based on its current susceptibility. To test this, for every sector $k$ in every country $c$ we consider its annual absolute output change, $Y_k^c(t+1) - Y_k^c(t)$, time-averaged over the range $t = 2000, \ldots, 2013$, $\langle \Delta Y_k^c \rangle_t = (1/13) \sum_{t=2000}^{2013} (Y_k^c(t+1) - Y_k^c(t))$. According to the above hypothesis, $\langle \Delta Y_k^c \rangle_t$ should be a function of susceptibility, $\rho_{ki}^c(t_0)$, and output, $Y_i(t_0)$, in the year $t_0 = 2000$. We therefore test the quality of the out-of-sample prediction given by

$$\langle \Delta Y_k^c \rangle_t = \eta \sum_i \rho_{ki}^c(t_0) Y_i(t_0). \tag{1}$$

Figure 2 shows that this relation indeed holds (Pearson's correlation coefficient of $r = 0.83$). This correlation is substantially stronger than the correlation between output change $\langle \Delta Y_k^c \rangle_t$ and output size $Y_k(t_0)$ alone ($r = 0.56$). Performing a linear regression of $\langle \Delta Y_k^c \rangle_t$ on $\sum_i \rho_{ki}^c(t_0) Y_i(t_0)$ and $Y_k(t_0)$ indeed yields a similar correlation as Eq. (1) alone (giving $r = 0.83$, with a regression coefficient of $-0.000(2)$ for $Y_k(t_0)$). Therefore, Eq. (1) adequately captures output fluctuations that go beyond trivial sector size effects. This confirms that the notion of economic susceptibility—the matrix $\rho_{ki}^c(t_0)$—coincides with (and is actually predictive of) the intuitive understanding that sectors with high susceptibility are those that are more easily moved by external events than low-susceptibility sectors.

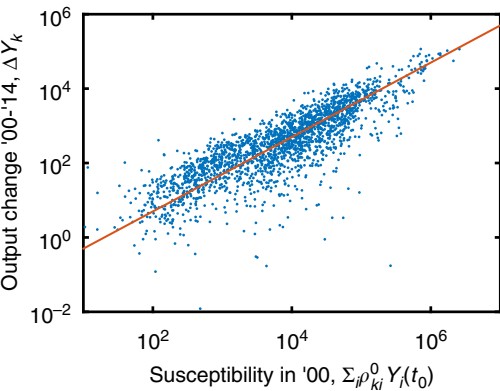

**Fig. 2** Prediction of output fluctuations with economic susceptibility. Under the assumption that each sector is driven by noise proportional to its output, we test the predictions that follow from the linear response framework, Eq. (1). We find good agreement between data and model ($r = 0.83$); economic susceptibility is indeed predictive of future output fluctuations. The red line has a slope of one, indicating a linear relation. Source data are provided as a Source Data file

We now show how the framework can be used to boost the quality of predictions of econometric timeseries models by extracting implied shocks from economic data. Finally, we illustrate potential applications of our results by discussing estimates for economic impacts of recent tariffs imposed on US–EU trades in steel and aluminum.

**Output predictions based on implied shocks.** The linear response formalism requires the specification of a demand shock in one or several sectors. Such shocks, however, can rarely be observed directly in the data. If a step demand shock occurs at the beginning of year $t$, the data from $t$ will not only contain the shock itself, but also of how the shock was digested by the economy during the year. As we have seen, recovery typically takes several years (see Fig. 1). However, one can compute implied shocks from the data as follows (for clarity we omit the country index $c$ from now on). Consider the truncated susceptibility matrix $\rho_{ik}(t, T)$, given by the area under the curve of the response function of $i$ to a shock in $k$, evaluated until $T$ years after the shock was applied (see Methods). Assume that changes in output between year $t$ and $t+1$ are due to a step demand shock $\tilde{X}_i(t') = \theta(t' - t)\tilde{X}_i$, with $\theta$ the Heaviside step function (see Methods). The size of this shock as implied by the output data from years $t$ and $t+1$ can be estimated by using Eq. (8),

$$\tilde{X}_i = (\rho(t, T=1))_{ik}^{-1}(Y_k(t+1) - Y_k(t)). \tag{2}$$

We refer to $\tilde{X}_i(t)$ as the implied shock at year $t$. Positive (negative) output changes typically coincide with implied shocks that are of even larger (smaller) value, though some sectors defy these general trends (see Supplementary Figure 3).

To test the validity of predictions of the linear response formalism, one can now take the implied shock from year $t$ and estimate the output in year $t+2$ using Eq. (8). Note that, by construction, the output in year $t+1$ is identical in the model and the data. This yields an LRT timeseries model for individual countries with a driven economy,

$$\left\langle Y_k^{\text{LRT}}(t+2) \right\rangle_X = Y_k(t) + \int_0^2 (\sigma^{-1})_{ij} \langle Y_k(t+\tau) Y_j(t) \rangle_0 \tilde{X}(t) d\tau. \tag{3}$$

The predictions of the LRT timeseries model are compared with expectations from econometric timeseries forecasting

methods, in particular to results from autoregressive integrated moving average (ARIMA) models[37] (see Supplementary Note 3 for a brief introduction). The respective performance of the ARIMA and LRT model is evaluated by Pearson's correlation coefficient between the actual (empirically observed) and predicted output changes. For each year $t$ and country $c$, we compute the correlation coefficient $r^{LRT}(c, t)$, between the empirical output, $Y_k(t)$, and the predictions from the LRT model $\left(Y_k^{LRT}(t)\right)$ in Eq. (3). Similarly, we compute the correlation coefficients $r^{ARIMA}(c, t)$ for predictions from the ARIMA model (correlation of $Y_k^{ARIMA}(t)$ with $Y_k(t)$). Values for $r^{LRT}(c, t)$ and $r^{ARIMA}(c, t)$ are shown in Supplementary Figures 4 and 5.

The differences between the correlation coefficients of two different models for the same country and year are referred to as the predictability gain, $PG(c, t) = r^{LRT}(c, t) - r^{ARIMA}(c, t)$ (see Fig. 3a). Red (blue) values indicate that for the given country and year the LRT model performs better (poorer) than the ARIMA model. For every year, we perform a $t$ test to reject the null hypothesis that the true mean of $PG(c, t)$, taken over all countries, is zero ($p < 0.05$). Figure 3b shows the $PG(c, t)$ averages over all countries taken at each year with a 95% confidence interval

(significant values are shown in black, nonsignificant in gray); Fig. 3c shows the results for every country (significant values are highlighted in black), and Fig. 3d shows the histogram of $PG(c, t)$ taken over all years and countries. The LRT model performs significantly better than the ARIMA model in almost each year and country. We find predictability gains of up to 100% and a $p$ value of $p < 10^{-46}$ to reject the null hypothesis that the true mean of the distribution of $PG(c, t)$ is zero in this timespan. Most intriguingly, for predictions from 2009 to 2010 (2 years after the crisis occurred) the LRT model shows by far the largest predictability gains. This result suggests that the LRT formalism works particularly well to describe the slow economic recovery during the Great Recession.

We design a further test, where it becomes harder for the LRT model to outcompete the ARIMA model, by comparing the out-of-sample predictions of the LRT model with the in-sample predictions of the ARIMA model. For this, we estimate the parameters of the sectoral ARIMA models over the entire timespan, from 2000 to 2014. This should clearly stack the deck against the LRT model, as the ARIMA model is now calibrated using full timeseries information, in particular on the speed of

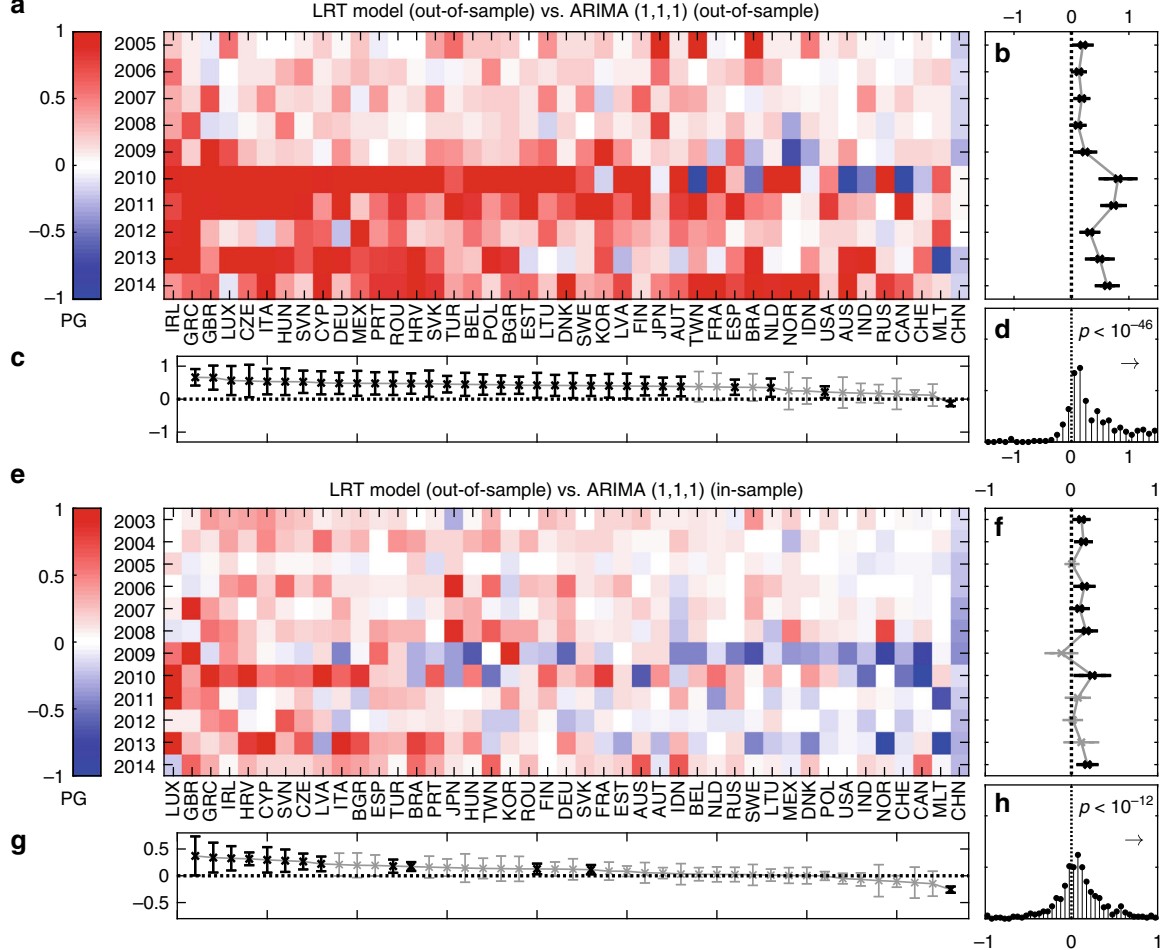

**Fig. 3** Comparison of the predictions of the linear response model with stochastic timeseries forecasting methods. **a** Comparison of the LRT model for a shock between years $t$ and $t + 1$ with an ARIMA(1,1,1) model that has been calibrated using data up to year $t + 1$. For every country and year, we show the predictability gain, PG of the LRT model over the ARIMA model, where **b** PG is averaged over all years or **c** over all countries. Averages that are significantly different from zero are highlighted. **d** The histogram of the PG over all countries and years shows the corresponding distribution. The LRT model drastically outperforms the ARIMA model, especially in the years that follow the crisis. The distribution of the predictability gains PG over all countries and years is significantly skewed towards positive values ($p < 10^{-46}$). **e–h** As in (**a–d**), however, the ARIMA model is calibrated by using the complete information of the entire timeseries. Its predictions are still outperformed by the LRT model ($p < 10^{-12}$). This means that the out-of-sample predictions of the LRT model are superior to in-sample (!) predictions from standard econometric forecasting models. Source data are provided as a Source Data file

economic recovery after the crisis. Results are shown in Fig. 3e–h. Overall, the LRT model again performs significantly better than the ARIMA model ($p < 10^{-12}$). The only exception is the prediction for 2009 where it is not clear which model is superior. In this case the ARIMA model had the chance to learn the speed of the autoregression directly after the crisis. In the following year, however, the LRT model shows the largest predictability gains, which again confirms that the LRT formalism is particularly useful to understand economic recovery. Given that the ARIMA model has access to the full information of the timeseries, whereas the predictions of the LRT model are always taken entirely out-of-sample, these results once more confirm the superiority of the LRT formalism in describing the response of economies to large recessionary shocks.

Here we showed results for the ARIMA(1, 1, 1) model. However, qualitatively the same results are obtained (in many cases with even stronger relative performance of the LRT model) for other types of model. In particular, in the Supplementary Information, we show results for the predictability gains $PG(c, t)$ for an ARIMA(1, 1, 0) model (differenced first-order autoregressive model), an ARIMA(0, 1, 1) model (exponential smoothing), and an ARIMA(1, 0, 1) model (first-order autoregressive moving average model) in Supplementary Figures 6–8. We also confirmed that the LRT model performs vastly superior to a sectoral VAR model (see Supplementary Note 4 and Supplementary Figure 9).

**Indirect effects of the US–EU trade war.** Finally, we show how the LRT model can be used to estimate the economic impact of instances such as the currently escalating trade war between the EU and US[38]. Starting from June 1, 2018, the US imposes a 25% tariff on steel and a 10% tariff on aluminum imports from member countries of the EU. These tariffs are expected to lead to direct negative effects on EU steel and aluminum producers, which could be further amplified by other countries that redirect their exports from the US to the EU. The indirect effects of these tariffs, however, are not so clear. Increased supply of steel and aluminum in the EU might lead to a decrease in price with positive effects on industries that require those metals as inputs. In the LRT model, the US tariffs impose a negative export demand shock on the manufacturing sector of basic metals (ISIC Code C24) on EU countries and a positive demand shock on the US. We assume that US demand in this category will reduce by 100% for European countries (and US domestic final consumption will increase accordingly) and estimate the resulting changes to the sectoral outputs using the linear relationship in Eq. (9). Note that the impacts of shocks with an arbitrary size of x% of current export demand can simply be estimated by multiplying these results by x/100. Results for $\langle \Delta Y_k \rangle_X$ obtained from Eq. (9) using the most recent data available in WIOD ($t = 2014$) are shown in Fig. 4a for the 25 largest sectors. In general, output changes fall in the range between ±0.5%. In European countries, positive effects are particularly strong in the manufacturing sectors (motor vehicles, computers, electronics, machinery, or electrical equipment), whereas there are consistently negative indirect effects for the energy sector. These findings are consistent with an expectation of positive effects further down the supply chain of steel and aluminum (due to price decreases). Decreases in the output of steel and aluminum production on the other hand coincide with a decrease in energy consumption. It is also apparent that the indirect effects are distributed nonuniformly across countries. Manufacturing activities in Germany, Greece, or Ireland show consistently increased levels of output. Indirect effects in the US often show opposite signs compared to the impact on European countries. We find that negative indirect

effects prevail for fabricated metal products and motor vehicles while the electricity sector, land transport, and wholesale trade experience positive effects. By summing the expected output changes (in USD terms) over each sector in a country, we obtain the aggregated indirect effects (Fig. 4b). Overall, almost all countries experience positive indirect effects with output increases of up to several billion dollars, the exceptions being Spain, Finland, Italy, and Romania. Our framework suggests that these countries might either depend to a higher extent on sectors that provide input to the manufacture of basic metals (such as electricity), lack sectors that can profit from an increased supply of basic metals, or that both of the former might be the case. Also, note that for European countries with positive aggregated indirect effects, these effects are typically outweighed by negative direct effects from the tariffs. Figure 4c shows the temporal impact (response curves) for Germany, for the a step demand shock for aluminum and steel.

## Discussion

We developed the theory of linear response for IO economies to quantify the resilience of national economies to production shocks. We established an analytic link between stationary output fluctuations and their out-of-equilibrium behavior. In particular, we derived the Green–Kubo relations for Leontief IO models, in full analogy to a wide range of physical phenomena, ranging from electrical and magnetic susceptibilities to shear viscosity and electrical resistance. Our framework can be applied to other types of IO model, as long as they are linear and permit a stationary solution. This includes IO models that use a higher geographic resolution (i.e. regional IO models), but also several of their generalizations, such as environmentally extended IO models[39], or commodity-by-industry IO models[40].

The central result of our work is a linear relationship between demand shock, $X_i$, and the induced output change $\Delta Y_k$, namely that $\langle \Delta Y_k \rangle_X = \rho_{ki} X_i$, with $\rho$ being a sector-by-sector matrix of economic susceptibilities. The output change $\langle \Delta Y_k \rangle_X$ characterizes a driven economy in a nonequilibrium stationary state. The original equilibrium state, $(\mathbb{I} - \mathbf{A})^{-1} \mathbf{D}$, is recovered for $X_i(t) = 0$ for all $i$ and $t$. The LRT solution $\langle \Delta Y_k \rangle_X$ is also fundamentally different from the perturbed equilibrium state implied by a step demand shock of the form $\mathbf{D_P} = \mathbf{D} + \mathbf{X}$ with $\mathbf{X}(t) = \mathbf{X}\theta(t)$, namely the perturbed equilibrium state $(\mathbb{I} - \mathbf{A})^{-1} \mathbf{D_P}$. To see the difference, note that in LRT the expectation values are taken over the probability density function of the stationary solution of $\dot{\mathbf{Y}} = (\mathbf{A} - \mathbb{I}) \mathbf{Y} + \mathbf{D} + \mathbf{F}(t)$ (from which the nonequilibrium expectation value $\langle \Delta Y_k \rangle_X$ is estimated), whereas the perturbed equilibrium state would be given by expectation values using the stationary solution of $\dot{\mathbf{Y}} = (\mathbf{A} - \mathbb{I}) \mathbf{Y} + \mathbf{D_P} + \mathbf{F}(t)$ as probability measure (see also Supplementary Note 5 and Supplementary Figure 10).

We demonstrated that our measures for economic susceptibility that can be derived from data are indeed predictive of future output fluctuations, even when no knowledge of future shocks is available. This finding corroborates that sectors with high susceptibility are indeed those that tend to be more easily movable by external shocks than low-susceptibility sectors. We showed that out-of-sample predictions from the LRT model consistently outperform standard econometric forecasting methods, such as different types of ARIMA model. Predictions of the LRT model work particularly well in the years that followed the recent financial crisis. This suggests that the LRT formalism allows us to get an analytic and quantitative understanding of the slow economic recovery of certain countries in the wake of the Great Recession. Because of the versatility and conceptual simplicity of input–output models, our framework can lead to more

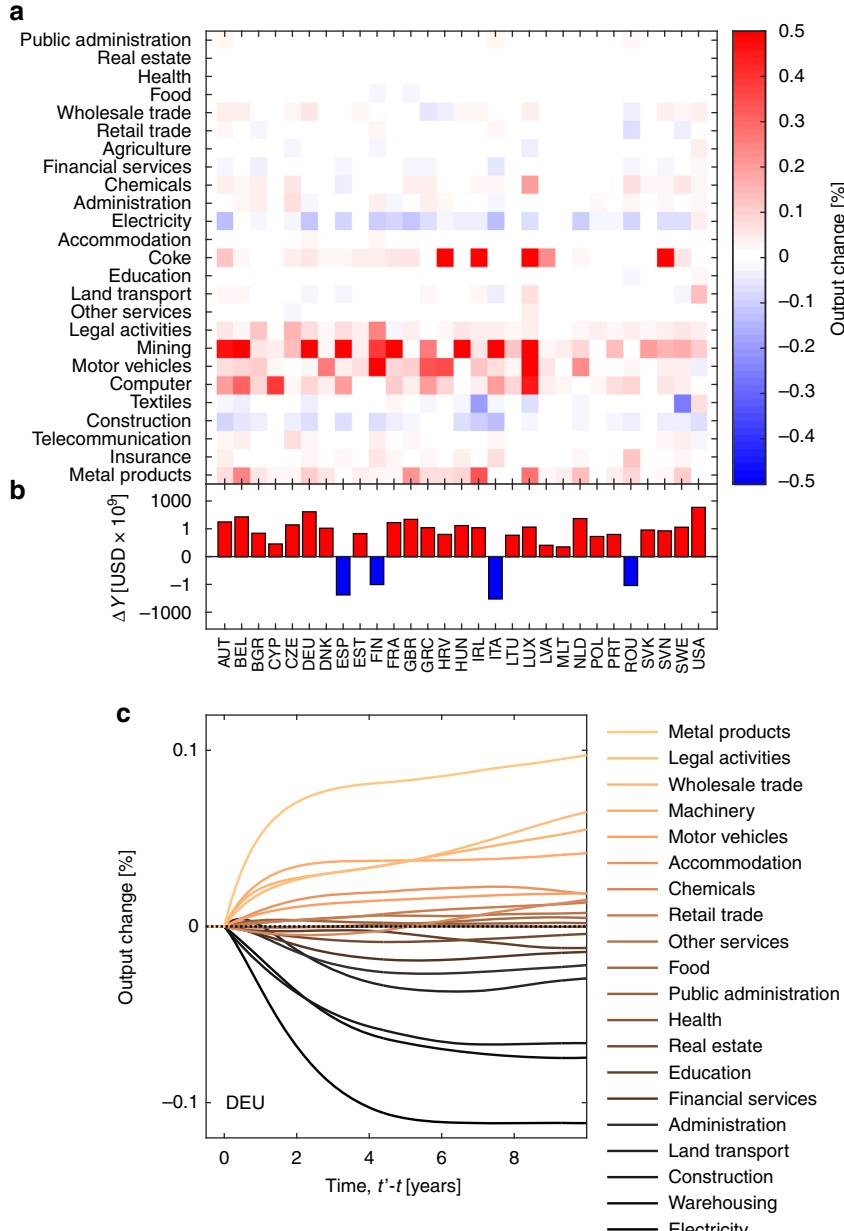

**Fig. 4** Estimation of indirect effects of the 2018 US steel and aluminum tariffs on EU countries. **a** For all sectors and countries we estimate output changes (in percent of 2014 outputs). Red (blue) colors indicate positive (negative) indirect effects. Sectors that require basic metals as input (e.g. the manufacture of motor vehicles or fabricated metal products) tend to show positive indirect effects in Europe; negative in the US. On the other hand, sectors as electricity or wholesale trade show mostly negative impacts in the EU and positive ones in the US. **b** For all countries we show the expected output change (in billion USD) due to indirect effects of the tariffs. Almost all countries experience positive indirect effects. Note that the y-axis scales logarithmically. **c** Response curves for Germany with a step demand shock in aluminum and steel. Source data are provided as a Source Data file

accurate quantitative estimates for the impact of disruptive events in various applications and scales, ranging from global recessions to regional, critical infrastructure systems. We illustrate the practical usefulness of our approach in providing concrete estimates for the indirect effects of the currently escalating US–EU trade war. In particular, we considered a negative export demand shock on the manufacturing sector of basic metals on EU countries and a corresponding positive demand shock on the US. We find that in European countries there is a trend toward positive indirect effects for manufacturing sectors further down the supply chain from basic metals, whereas electricity outputs show negative indirect effects. In the US we find similar results with

reversed signs; positive (negative) effects moving further up (down) along the supply chain.

A limitation of the Leontief IO model that extends to our work is that prices play no role in the model. Firms in real economies can respond to shocks by adjusting produced quantities as well as prices. It therefore remains to be seen how prices can be incorporated in the LRT framework, i.e. within a linear time-invariant formulation of the underlying microscopic dynamics. Besides linearity and time-invariance, our approach also assumes an external shock that may depend only on time and for which we only consider first-order correction with respect to the unperturbed state.

In summary, in this work we extended current mainstream economic theories to out-of-equilibrium situations in a way that is analytically rigorous, empirically testable, and flexible enough to immediately address a wide range of scenarios with a direct political relevance, such as identifying those parts of a country's economy that are particularly vulnerable in a trade war.

## Methods

**Linear response theory of input–output economics.** Consider an economy with $N$ sectors, each sector producing $Y_i$ units of a single homogeneous good. Assume that sector $j$ requires $A_{ij}$ units from sector $i$ as input to produce one unit itself, which gives the so-called technical coefficients $A_{ij}$. Each sector sells some of its output to consumers, the demand $D_i$. The open Leontief IO model, the standard model in economics to depict and analyze inter-sectoral relationships, assumes linear production functions given by $\mathbf{Y} = \mathbf{A}\mathbf{Y} + \mathbf{D}$ (matrix notation). The stationary (equilibrium) state of this economy is given by $\mathbf{Y}^0 = (\mathbb{I} - \mathbf{A})^{-1}\mathbf{D}$ ($\mathbb{I}$ being the $N$-dimensional identity matrix). For the time evolution of an economy in its stationary state, assuming that differences in dynamic demand $\mathbf{A}\mathbf{Y} + \mathbf{D}$ and dynamic production $\mathbf{Y}$ are compensated by production changes, this model gives the differential equation[33]

$$\dot{\mathbf{Y}} = (\mathbf{A} - \mathbb{I})\mathbf{Y} + \mathbf{D}. \tag{4}$$

We assume that each sector $i$ experiences a time-dependent demand shock, $X_i(t)$, and the presence of multivariate white noise, i.e. a stochastic force, $F_i(t)$. In the picture of the example of the house of cards given above, the noise $F_i(t)$ represents the tiny nudge that we apply to the serving tray to understand if the house would survive a much larger shock, $X_i(t)$. More formally, the nudge consists of noise with mean value $\langle F_i(t)\rangle_0 = 0$, and covariance $\langle F_i(t)F_j(s)\rangle_0 = v_{ij}\delta(t - s)$. Here, $\delta(x)$ denotes the Dirac-delta function, $v$ is a matrix of constants, and $\langle \mathbf{x}(t)\rangle_0 = \int d^N\mathbf{Y}\mathbf{x}(\mathbf{Y})f_0(\mathbf{Y})$ is the equilibrium expectation value of the function $\mathbf{x}(t)$, evaluated in the absence of an external force ($\mathbf{X}(t) = 0$), with $f_0(\mathbf{Y})$ being the probability distribution to find a given value of $\mathbf{Y}$ under noise $F_i(t)$. This leads to the stochastic differential equation,

$$\dot{\mathbf{Y}} = (\mathbf{A} - \mathbb{I})\mathbf{Y} + \mathbf{D} + \mathbf{X}(t) + \mathbf{F}(t). \tag{5}$$

From the central limit theorem it follows immediately that the stationary or equilibrium solution $f_0(\mathbf{Y})$ in the absence of external shocks ($\mathbf{X}(t) = 0$) of Eq. (5) is given by a multivariate normal distribution with covariance $\sigma_{ij} = \lim_{t\to\infty}\langle Y_i(t) Y_j(t)\rangle_0$. In the presence of external shocks, i.e. for $\mathbf{X}(t) \neq 0$, a solution of Eq. (5) with first-order corrections from the shock can be obtained using LRT (see Supplementary Note 6). We denote the expectation value for the output change of sector $k$ with nonzero shock $\mathbf{X}(t)$ by $\langle \Delta Y_k(t)\rangle_{\mathrm{X}} \equiv \langle Y_k(t)\rangle_{\mathrm{X}} - Y_k^0$. That is, averages with a subscript 0 refer to values taken at equilibrium, whereas averages with a subscript X refer to out-of-equilibrium properties. Following LRT[32], we get the general solution for the time evolution of the output changes, $\langle \Delta Y_k(t)\rangle_{\mathrm{X}}$,

$$\langle \Delta Y_k(t)\rangle_{\mathrm{X}} = \int_{-\infty}^{t}(\sigma^{-1})_{ij}\left\langle Y_k(\tau)Y_j(0)\right\rangle_0 X_i(\tau)\mathrm{d}\tau. \tag{6}$$

Remarkably, we have related the out-of-equilibrium response of the sectoral outputs, $\langle \Delta Y_k(t)\rangle_{\mathrm{X}}$, to their correlation functions taken at equilibrium. Equation 6 characterizes the state of a driven economy.

For certain types of demand shock, the resulting output change takes a particularly simple form. For an impulse demand shock, $X_i(t) = \delta(t)X_i$, we get

$$\left\langle \Delta Y_k^{\mathrm{pulse}}(t)\right\rangle_{\mathrm{X}} = (\sigma^{-1})_{ij}\langle Y_k(t)Y_j(0)\rangle_0 X_i. \tag{7}$$

For a step demand shock, $X_i(t) = \theta(t)X_i$ with the Heaviside step function $\theta(t \geq 0) = 1$ and $\theta(t < 0) = 0$, we get

$$\left\langle \Delta Y_k^{\mathrm{step}}(t)\right\rangle_{\mathrm{X}} = \int_0^t(\sigma^{-1})_{ij}\langle Y_k(\tau)Y_j(0)\rangle_0 X_i\mathrm{d}\tau. \tag{8}$$

For $t \gg 0$ we obtain the linear relation

$$\langle \Delta Y_k\rangle_{\mathrm{X}} = \rho_{ki}X_i, \text{ with } \rho_{ki} = \int_0^\infty(\sigma^{-1})_{ij}\langle Y_k(\tau)Y_j(0)\rangle_0\mathrm{d}\tau, \tag{9}$$

where we introduced the economic susceptibility $\rho$, in full analogy to the derivation of electric or magnetic susceptibilities in statistical mechanics (see Supplementary Table 1). The economic susceptibility $\rho_{ki}$ has the precise meaning of output change in sector $k$, given that a step demand shock of unit size occurs in sector $i$.

In this paper we encounter different types of susceptibility, depending on how averages are taken. In particular we will use the following definitions: The $N \times N$ susceptibility matrix of a country $c$ at year $t$ is defined by

$$\rho_{ij}^c(t) = \int_0^\infty(\sigma^{-1})_{ij}\langle Y_k^c(t + \tau)Y_j^c(t)\rangle_0\mathrm{d}\tau, \tag{10}$$

where the output $Y_k^c(t)$, technical coefficients $A_{ij}^c(t)$, and the demand $D_i^c(t)$ of a particular country $c$, are read off the WIOD[34]. A truncated version $\rho_{ij}^c(t, T)$ of this susceptibility matrix is obtained by taking $t + T$, $T > 0$, as the upper boundary of the integration range in Eq. (10). The susceptibility of sector $i$ in country $c$ at year $t$,

$\rho_i^c(t)$, is defined as the corresponding column sum of the susceptibility matrix, $\rho_i^c(t) = \sum_j\rho_{ij}^c(t)$. We define the averaged country susceptibility as the average of the sector susceptibility taken over all $N$ sectors and $N_t = 15$ years, $\rho^c = (N_tN)^{-1}\sum_{i,t}\rho_i^c(t)$. The output-weighted average sector susceptibility, $\rho_i$ is defined as, $\rho_i = (N_tN_c\sum_{t,c}Y_i^c(t))^{-1}\sum_{t,c}Y_i^c(t)\rho_i^c(t)$, where $N_c = 43$ is the number of countries in the data.

## Data availability

The study is based on the 2016 release of the World Input–Output Tables[34] (see also http://www.wiod.orghttp://www.wiod.org, accessed 15 January 2019). The source data underlying all main and supplementary figures are provided as a Source Data file.

## Code availability

Code is available upon request directly from the authors.

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

## Acknowledgements

We thank M. Miess and A. Pichler for helpful discussions, J. Sorger for help with the visualizations, and acknowledge support from the European Commission, H2020 SmartResilience No. 700621, FFG Project 857136, and OeNB Jubiläumsfond project 17795.

## Author contributions

P.K. and S.T. designed research, P.K. performed research and analyzed data, S.P. contributed timeseries models, P.K. and S.T. wrote the paper.

## Additional information

**Competing interests:** The authors declare no competing interests.

