## [Peer Review File · Nature Communications]

Reviewers' comments:

Reviewer #1 (Remarks to the Author):

This is a well written and informative technical paper on the applications of input-output models for quantitative economic analysis. The multiple authors from several countries provide unique global perspectives on economic resilience and economic growth.

The authors make measurable contributions to our understanding of economic resilience--the ability to recover from adverse initiating events--within acceptable time frame and consequences.

The reviewer finds the "Supporting Information: Texts S1-S4 to be essential to the paper and must accompany the published document.

Reviewer #2 (Remarks to the Author):

In this manuscript, the authors propose to study how economies react to shocks using Linear Response Theory (LRT), a statistical-physics scheme that relates how a system behaves after a perturbation that drives it out of equilibrium to its random fluctuations in equilibrium. They show that their methodology can be fitted to real data and outperforms other currently used models in predicting the evolution of real economic systems. In addition to the predictive power of their model, the approach also enables two interesting possibilities. On the one hand, their analysis allows to compute the susceptibilities, which encode the interdependencies among the different sectors of a given economy by quantifying how each of them affects the others when perturbed. On the other hand, their mathematical approach can be used to infer shocks implied by the observed data. In my opinion, this is a very interesting contribution that seems to have the potential to be used in real economic scenarios. Moreover, I think that the authors have done a very good job at presenting the topic to a wide audience in the main text.

In general, I like the manuscript, but I think there are a few aspects that should be presented more clearly before publication in Nature Communications:

- Fig. 1C depicts the evolution of the susceptibilities of groups of sectors along with their corresponding response to a delta-like shock. However, it is quite difficult to appreciate sectors' strengths (degrees) in the backbones and also the changes in nodes' sizes. Therefore, the correlation

between these two quantities is not obvious at all. I recommend to find some other way to represent this.

· At the end of the first paragraph in the discussion, the authors affirm that the state of a driven economy for which $X \neq 0$ "is different from [...] an equilibrium state that would be implied by, for instance, a perturbed demand of the form $D' = D + X$ ". I think this point requires further clarification. In particular, if the system is governed by Eq. (4), what is the fundamental difference between the stationary solution after a step demand shock as $t \rightarrow \infty$ and a perturbed demand $D' = D + X$? Should not both solutions satisfy the same equation $\dot{Y} = 0 = (A - I)Y + D + X$? In other words, does the system not approach the stationary solution $Y^0 = (I - A)^{-1}(D + X)$, which corresponds to the equilibrium solution for $D' = D + X$ (for which there would also be a response of the form of Eq. (8), $\Delta Y = (I - A)^{-1}X$), as $t \rightarrow \infty$ after a step shock?

· The derivation in Supporting Text S6 is somewhat confusing. In Eq. (12), for instance, $L_X(y, t)$ is used on both sides of the equation, which makes unclear to which of them the authors refer thereon (like in Eqs. (13) and (15)-(17)). Also, I think the exposition would benefit from some comments regarding the assumptions and corresponding approximations taken in the derivation.

· This last point is just a question, so perhaps no further clarification is required in the manuscript. To obtain the time-lagged correlation functions, Eq. (4) is numerically integrated with $X = 0$. In terms of predictive power of the approach, would integrating Eq. (4) numerically again once the implied shock has been inferred instead of using Eq. (2) improve the results?

Finally, there is also a few minor details:

· In page 2, first paragraph of the Results section, it should be stated that the subscripts in the demands, technical coefficients and outputs refer to sectors.

· In page 7, 7th line below Eq. (4), I think the reference to SI Text S5 should be S6.

· Two citations are missing in the Supporting Information (second line in Text S2 and 4th line after Eq. (11)).

Reviewer #3 (Remarks to the Author):

I read with interest the paper “Economic resilience from input–output susceptibility improves predictions of economic growth and recovery” by P. Klimek et al.

The paper applies methodology of Statistical Physics to the analysis and forecasting of economic systems, in particular the study of the recovery after a crisis. In order to do so the authors describe the economic system by means of input-output data from industrial sectors. The paper is certainly right in pointing out that standard economic theory (to the best of my knowledge as a physicist) scarcely deals with out-of-equilibrium situations (where on the other hand also standard statistical physics has some problems). Authors therefore apply Linear Response Theory to describe the evolution of the system after the initial shock.

One point that needs to be clarified is the use of central limit theorem (see e.g. eq.4) and in general LRT for economic systems that are known to be fat-tailed distributed. Authors should clarify the limits in which their approach is valid.

Minor points

Problems in the url of ref. 38

As for the last sentence of Text S1, I would even say that thermodynamics is not defined out of equilibrium.

Some of the references are missing in all the SI part (a “?” appears)

The format to introduce Laplace transform properties (between eqs 17 and 18 of SI) is rather obscure, since we are in the SI part, please expand it and comment or remove it.

In Text S6 “The application to stochastic processes” The central limit theorem might be used only if the variance of the $y_i(t)$ and $y_j(t)$ is finite. Economic and financial systems are often characterized by fat-tailed (sometimes power law distribution

We want to thank all three referees for their time and effort that went into reviewing our work. We are extremely glad to see that each of them finds merit in our approach and provide us with valuable constructive feedback. We have taken each of these comments thoroughly into account and changed the manuscript accordingly. In particular, we adjusted the length of the manuscript, clarified some of the non-equilibrium properties of our approach, improved the presentation of the derivation, and added a more thorough discussion of the necessary assumptions of our method, in particular in relation to the use of the Central Limit Theorem (CLT).

Reviewers' comments:

Reviewer #1 (Remarks to the Author):

This is a well written and informative technical paper on the applications of input-output models for quantitative economic analysis. The multiple authors from several countries provide unique global perspectives on economic resilience and economic growth.

The authors make measurable contributions to our understanding of economic resilience--the ability to recover from adverse initiating events--within acceptable time frame and consequences.

The reviewer finds the "Supporting Information: Texts S1-S4 to be essential to the paper and must accompany the published document.

It is a pleasure to read that the reviewer finds our paper to be “well written and informative”. Concerning the suggestion to move Texts S1-S4 to the main text: unfortunately, we cannot fully comply with this suggestion due to length constraints. In fact, according to journal policies the length of the introduction should be less than 1,000 words, and our original introduction had about 1,200 words, so we had to adapt the introduction to a more concise form. We could, however, move Text S3 to the main text while still being able to comply with length constraints.

Reviewer #2 (Remarks to the Author):

In this manuscript, the authors propose to study how economies react to shocks using Linear Response Theory (LRT), a statistical-physics scheme that relates how a system behaves after a perturbation that drives it out of equilibrium to its random fluctuations in equilibrium. They show that their methodology can be fitted to real data and outperforms other currently used models in predicting the evolution of real economic systems. In addition to the predictive power of their model, the approach also enables two interesting possibilities. On the one hand, their analysis allows to compute the susceptibilities, which encode the interdependencies among the different sectors of a given economy by quantifying how each of them affects the others when perturbed. On the other hand, their mathematical approach can be used to infer shocks implied by the observed data. In my opinion, this is a very interesting contribution that seems to have the potential to be used in real economic scenarios.

Moreover, I think that the authors have done a very good job at presenting the topic to a wide audience in the main text.

In general, I like the manuscript, but I think there are a few aspects that should be presented more clearly before publication in Nature Communications:

· Fig. 1C depicts the evolution of the susceptibilities of groups of sectors along with their corresponding response to a delta-like shock. However, it is quite difficult to appreciate sectors' strengths (degrees) in the backbones and also the changes in nodes' sizes. Therefore, the correlation between these two quantities is not obvious at all. I recommend to find some other way to represent this.

First, we also want to express our gratitude toward reviewer 2 for the helpful feedback. In response to the above comment, we have changed Fig. 1C to also visually encode the magnitude of the change and the node strengths in the network. We hope that this allows the reader to gain a better intuition on the structure that drives the observed response curves. A full formal understanding of how the response curves relate to entries in the susceptibility matrix can be gained from the analytical derivation of the response curves in addition to this visualization.

· At the end of the first paragraph in the discussion, the authors affirm that the state of a driven economy for which $X \neq 0$ "is different from [...] an equilibrium state that would be implied by, for instance, a perturbed demand of the form $D' = D + X$ ". I think this point requires further clarification. In particular, if the system is governed by Eq. (4), what is the fundamental difference between the stationary solution after a step demand shock as $t \rightarrow$ infinity and a perturbed demand $D' = D + X$? Should not both solutions satisfy the same equation $\dot{Y} = 0 = (A - I)Y + D + X$? In other words, does the system not approach the stationary solution $Y^0 = (I - A)^{-1}(D + X)$, which corresponds to the equilibrium solution for $D' = D + X$ (for which there would also be a response of the form of Eq. (8), $\Delta Y = (I - A)^{-1}X$), as $t \rightarrow$ infinity after a step shock?

The "driven economy" refers to the vector of outputs being in a stationary state that is different from the equilibrium state implied by the (perturbed or unperturbed) Leontief model. To see this on a more formal level, note that the LRT result for the asymptotic output change for a step-demand shock of the form $D+X$ (as described above) would be $\Delta X = \rho X$, with ρ being the country's susceptibility matrix. To see that ρ must not be equal to the Leontief inverse $(I - A)^{-1}$, note that we can write the Leontief inverse as a power series of the form $\sum_{k=0}^{\infty} A^k$. As all entries in A are positive semidefinite, all entries in A^k for $k=0,1,2,\dots$ must also be positive semidefinite. However, as shown in the paper, entries in the susceptibility matrix can also be (and a substantial number of entries are) negative.

To answer this question more physically, the stationary non-equilibrium LRT state is characterized through a stationary transport process that counters an externally imposed gradient, whereas there is no such gradient in the stationary equilibrium state.

Note that the fact that entries in ρX can be negative, whereas entries in $(I-A)^{-1}X$ are always non-negative, has far-reaching consequences when we consider the example of the "Trump tax", i.e. the alu and steel tariffs. If we were to estimate the effects of a demand reduction on other sectors using the Leontief inverse for shock propagation, we would necessarily conclude that all indirect effects on other sectors are either zero or negative – but there can be no "winners", i.e. sectors that profit from the reduction of demand in another sector. In our approach, as we show, this is very well the case. So, in response to this comment, we have added a paragraph in the discussion section when referring to the indirect effects of the steel tariffs where we state that the observed positive indirect effects are an effect from our non-equilibrium formalism that could not be accounted for in the standard Leontief model.

· The derivation in Supporting Text S6 is somewhat confusing. In Eq. (12), for instance, $L_X(y, t)$ is used on both sides of the equation, which makes unclear to which of them the authors refer thereon (like in Eqs. (13) and (15)-(17)). Also, I think the exposition would benefit from some comments regarding the assumptions and corresponding approximations taken in the derivation.

Thank you for making us aware of these inaccuracies. We introduced a different symbol, $\Lambda_X(y, t)$ for the evolution operator. Also, we have added an explicit listing of all assumptions and approximations that have been made in the last paragraph of the supporting text element, namely that the dynamical system is (i) linear, (ii) time invariant with an external perturbation that (iii) depends only on time. In addition, we only consider first order effects of this perturbation and neglect non-linear, higher order effects.

· This last point is just a question, so perhaps no further clarification is required in the manuscript. To obtain the time-lagged correlation functions, Eq. (4) is numerically integrated with $X = 0$. In terms of predictive power of the approach, would integrating Eq. (4) numerically again once the implied shock has been inferred instead of using Eq. (2) improve the results?

That's an interesting question that we have not considered before. Note that we integrate Eq. (4), as is stated in the text, in the absence of an external shock, $X(t) = 0$. The reason why we didn't consider different scenarios is because it is not at all clear how an expectation value of the correlation function can be defined, let alone computed, if we leave the analytical framework of LRT and try to consider expectation values in a non-stationary regime. As our manuscript propagates an analytical rigorous approach, we think that an exploration of this question would go beyond the scope of the current paper.

However, what the referee proposes sounds like the idea behind non-linear response. In this extension of LRT, one considers a thermostatted system, for instance a fluid that is sheared and elongated while keeping the kinetic energy in the ensemble fixed by introducing an additional coupling term in the equations of motion. Such systems have been shown to possess nonlinear transport coefficients that can be expressed in terms of so-called transient time correlation functions, i.e. correlations functions that are computed under the perturbed equations of motions (with the perturbation acting as a thermostat for one of the dynamic variables). It could be worthwhile to think about what this thermostatted variable would correspond to in terms of economic production processes in order to study nonlinear response theory, but at the current stage this would all be purely speculative...

Finally, there is also a few minor details:

· In page 2, first paragraph of the Results section, it should be stated that the subscripts in the demands, technical coefficients and outputs refer to sectors.

Thank you, has been added

· In page 7, 7th line below Eq. (4), I think the reference to SI Text S5 should be S6.

Indeed, was corrected.

· Two citations are missing in the Supporting Information (second line in Text S2 and 4th line after Eq. (11)).

Thank you again, something went wrong with formatting URLs...

Reviewer #3 (Remarks to the Author):

I read with interest the paper “Economic resilience from input–output susceptibility improves predictions of economic growth and recovery” by P. Klimek et al.

The paper applies methodology of Statistical Physics to the analysis and forecasting of economic systems, in particular the study of the recovery after a crisis. In order to do so the authors describe the economic system by means of input-output data from industrial sectors. The paper is certainly right in pointing out that standard economic theory (to the best of my knowledge as a physicist) scarcely deals with out-of-equilibrium situations (where on the other hand also standard statistical physics has some problems). Authors therefore apply Linear Response Theory to describe the evolution of the system after the initial shock. One point that needs to be clarified is the use of central limit theorem (see e.g. eq.4) and in general LRT for economic systems that are known to be fat-tailed distributed. Authors should clarify the limits in which their approach is valid.

We thank the referee for the interest in our work and the help to further improve its quality. With respect to the limits of the validity of our approach we have added an explicit summary of all assumptions and approximations in the discussion. From these assumptions it follows also that in the absence of external shocks changes in output would follow the CLT. Note that this is not at all at variance with the empirical observation of fat-tailed distributions in output changes, as we make no statement whatsoever about the distribution of output changes in the non-equilibrium regime, see below.

Minor points

Problems in the url of ref. 38

Thank you, problem fixed.

As for the last sentence of Text S1, I would even say that thermodynamics is not defined out of equilibrium.

Indeed, we have added a statement that it is questionable to talk about thermodynamics in a non-equilibrium regime at all.

Some of the references are missing in all the SI part (a “?” appears)

The references have been fixed.

The format to introduce Laplace transform properties (between eqs 17 and 18 of SI) is rather obscure, since we are in the SI part, please expand it and comment or remove it.

We have rewritten and expanded this part to make the steps that involve the Laplace transform more transparent and easier to follow.

In Text S6 "The application to stochastic processes" The central limit theorem might be used only if the variance of the $y_i(t)$ and $y_j(t)$ is finite. Economic and financial systems are often characterized by fat-tailed (sometimes power law distribution

We have clarified that the variance refers to the variance measured at equilibrium, i.e. in the stationary state, and therefore all the expectation values that are involved in the computation of the variance are equilibrium expectation values, i.e. computed in the absence of an external shock. We have also added a brief discussion of the use of the CLT in Text S6 where we state that there is no problem with the fact that many time series show fat-tailed distributions as the stationary solution cannot be directly observed (due to the ongoing presence of direct or indirect impacts of shocks).

Reviewers' comments:

Reviewer #2 (Remarks to the Author):

The authors have corrected the minor points raised in my previous report, but I am afraid that I cannot agree with their argument regarding the impossibility for $(I - A)^{-1}X$ to yield positive indirect effects for a reduction shock. They claim that $(I - A)^{-1}$ can be expanded as the series $\sum_k A^k$ and, since all the elements of A are greater than or equal to zero, so must be the elements of $(I - A)^{-1}$. Consequently, $(I - A)^{-1}X$ cannot have any positive components if X represents a negative shock. However, the series expansion proposed is not always convergent and, as a result, it is not always valid. For instance, for $A = \begin{pmatrix} 0 & 2 \\ 2 & 0 \end{pmatrix}$, all entries of $(I - A)^{-1}$ are negative. In fact, the necessary and sufficient condition for matrix A to allow for an equilibrium solution (that is, for which both Y and D satisfying $\dot{Y} = 0$ are positive semi-definite), known as the Hawkins-Simons condition, is that all the principal minors of $I - A$ are positive (see for instance "Mathematics for Stability and Optimization of Economic Systems" by Y. Murata, Theorem 30). As it turns out, the data used by the authors does not fulfill this condition in general. In particular, the National IO Table for Germany in 2014 from the WIOD yields a matrix $(I - A)^{-1}$ which contains both positive and negative values. What is more, setting X to have a unique non-null negative value for the "Manufacture of basic metals" component yields a vector $(I - A)^{-1}X$ whose largest component, which is positive, is "Mining and quarrying", in accordance with Fig 4A. Of course, the resulting vector also has negative components. I might have misunderstood some important point or made some mistake, and I will be happy to be proven wrong, but I believe that, in light of this, the need to use LRT should be further justified. More precisely, the following questions should be addressed:

1. Is the susceptibility matrix proposed in this work fundamentally different from $(I - A)^{-1}$ (beyond the approximations followed to derive them). If so, which yields the best predictions? So far, the authors have assumed that $(I - A)^{-1}$ could not yield indirect effects of different sign, so its performance has not been considered.

2. In my previous report, I asked about the stationary solution found by LRT, but the authors' response was not clear in that respect, so I am still confused about the distinction between equilibrium and stationary solutions in this particular scenario, where both cases are driven by the same dynamical equation. Let me rephrase my question more precisely. In Eq. (9), the authors find a linear relation between ΔY and X , which they claim to be different from the perturbed equilibrium solution. They call the final state in this case a stationary state. As I previously stated, if stationary here means $\langle \dot{Y} \rangle = 0$, then the solution must be $\Delta Y = (I - A)^{-1}X$. However, I understand that if that result is different from ρX , then the stationary solution does not fulfill the dynamical equation with $\langle \dot{Y} \rangle = 0$. Does this mean that the long-time solution of the dynamical equation is oscillating if a step shock is assumed? In that case, does ΔY represent a temporal average? I would appreciate some further explanation.

Reviewer #3 (Remarks to the Author):

I found the new version of the paper substantially improved and I suggest publication in the present form

1 Reviewer comment

The authors have corrected the minor points raised in my previous report, but I am afraid that I cannot agree with their argument regarding the impossibility for $(I - A)^{-1}X$ to

yield positive indirect effects for a reduction shock. They claim that $(I - A)^{-1}$ can be expanded as the series $\sum A^k$ and, since all the elements of A are greater than or equal to zero, so must be the elements of $(I - A)^{-1}$. Consequently, $(I - A)^{-1}X$ cannot have any positive components if X represents a negative shock. However, the series expansion proposed is not always convergent and, as a result, it is not always valid. For instance, for $A = \begin{pmatrix} 0 & 2 \\ 2 & 0 \end{pmatrix}$, all entries of $(I - A)^{-1}$ are negative. In fact, the necessary and sufficient condition for matrix A to allow for an equilibrium solution (that is, for which both Y and D satisfying $\dot{Y} = 0$ are positive semi-definite), known as the Hawkins-Simons condition, is that all the principal minors of $I - A$ are positive (see for instance "Mathematics for Stability and Optimization of Economic Systems" by Y. Murata, Theorem 30). As it turns out, the data used by the authors does not fulfill this condition in general. In particular, the National IO Table for Germany in 2014 from the WIOD yields a matrix $(I - A)^{-1}$ which contains both positive and negative values. What is more, setting X to have a unique non-null negative value for the "Manufacture of basic metals" component yields a vector $(I - A)^{-1}X$ whose largest component, which is positive, is "Mining and quarrying", in accordance with Fig 4A. Of course, the resulting vector also has negative components. I might have misunderstood some important point or made some mistake, and I will be happy to be proven wrong, but I believe that, in light of this, the need to use LRT should be further justified. More precisely, the following questions should be addressed:

1.1 Issue 1

1. Is the susceptibility matrix proposed in this work fundamentally different from $(I - A)^{-1}$ (beyond the approximations followed to derive them). If so, which yields the best predictions? So far, the authors have assumed that $(I - A)^{-1}$ could not yield indirect effects of different sign, so its performance has not been considered.

1.2 Issue 2

2. In my previous report, I asked about the stationary solution found by LRT, but the authors' response was not clear in that respect, so I am still confused about the distinction between equilibrium and stationary solutions in this particular scenario, where both cases are driven by the same dynamical equation. Let me rephrase my question more precisely. In Eq. (9), the authors find a linear relation between ΔY and X , which they claim to be different from the perturbed equilibrium solution. They call the final state in this case a stationary state. As I previously stated, if stationary here means $\langle \dot{Y} \rangle = 0$, then the solution must be $\Delta Y = (I - A)^{-1}X$. However, I understand that if that result is different from ρX , then the stationary solution does not fulfill the dynamical equation with $\langle \dot{Y} \rangle = 0$. Does this mean that the long-time solution of the dynamical equation is oscillating if a step shock is assumed? In that case, does ΔY represent a temporal average? I would appreciate some further explanation.

2 Our response

We thank the reviewer for pointing out an inaccuracy in our prior argument concerning the necessary validity of the Hawkins-Simon condition. In response to these comments, we have made several changes to make the fundamental difference between the solution provided by the Leontief inverse matrix and the LRT results clearer. We have reworked parts of the discussion to provide more space to explain the conceptual difference between the non-equilibrium and equilibrium stationary solutions. In addition, we have added a supplementary note that addresses this point in more technical depths. In the following we respond to the two points raised above.

2.1 Response to issue 1

The fundamental difference between the susceptibility matrix and the Leontief inverse is that the susceptibility matrix (and as such the LRT prediction for the output change) is obtained as an expectation value that uses a different probability measure (namely one associated with the stationary solution of the unperturbed IO model) compared to the derivation of the equilibrium expectation value of the "perturbed" IO system, where the stationary solution turns indeed out to be a multivariate normal distribution centred on the vector $(I - A)^{-1}X$. We seek to fully clarify this point in our response to the second issue.

Let us turn to the question of which model provides better predictions. A direct comparison of those two models (LRT and "perturbed" Leontief IO model) as indicated by the referee is shown in figure 1 which shows that LRT predictions clearly perform better ($p < 10^{-90}$ following exactly the same evaluation procedure as described in the manuscript for the other models). Note that we considered adding this analysis to the manuscript when we were working on early drafts. However, we decided against using this for the following reason: The analysis can only be performed when the shock X is known. LRT provides us with a way to estimate X based on past observations, the

Figure 1: Comparison of the predictions of the linear response model with a perturbed Leontief IO model as suggested by the referee. Overall, predictions of the LRT model clearly outperform the deterministic Leontief IO model with perturbed demand.

implied shock. Now using this LRT-solution (the implied shock) as demand shock in the Leontief IO model would be inconsistent, as we would effectively describe the state of the economy using the LRT model backward in time and the standard Leontief model forward in time. After all, we can compute the impact of the shock using the LRT results directly! So such a comparison would violate the analytical rigour that we propagate in the abstract. As we have added a new supplementary note now that discusses the differences between these models in greater detail, we hope that it is not too distracting and misleading to add this plot as another supplementary figure. However, we propose to keep it “detached” from the other comparisons (timeseries models) for the reasons outlined above.

2.2 Response to issue 2

We thank the referee again for making us aware that this point needs further elaboration to work out the difference between equilibrium and non-equilibrium expectation values. We have added a supplementary note in the manuscript that clarifies their differences. We have also reworked the discussion section in order to give a condensed version of this argument; for the full technical details the reader is then referred to the supplement. Also, we noted that at some places in the discussion section of the original manuscript we dropped the expectation value notation in the hope of getting a crisper presentation

of the results. As this might have additionally confounded the misunderstandings concerning which expectation value to use, we have also corrected these occurrences. In the following, and for ease of reference, we give now the supplementary note that has been added to discuss the issue raised by the referee:

In this note we clarify the difference between the susceptibility matrix ρ and the Leontief inverse matrix. In particular, for a step demand shock in LRT we find a non-equilibrium stationary state given by the linear relation $\langle \Delta \mathbf{Y} \rangle_X = \rho \mathbf{X}$, whereas in the “standard” Leontief IO economy we would expect a perturbed equilibrium state given by $\Delta \mathbf{Y} = (\mathbb{I} - \mathbf{A})^{-1} \mathbf{X}$. Wherein lies the fundamental difference between these two expressions?

To obtain the LRT solution for the output change, we describe the Leontief IO model in terms of a stochastic differential equation, see Eq. (5) in the main text,

$$\dot{\mathbf{Y}} = (\mathbf{A} - \mathbb{I})\mathbf{Y} + \mathbf{D} + \mathbf{X}(t) + \mathbf{F}(t) \quad . \quad (1)$$

The formal and stationary solution of this equation in the absence of a shock ($\mathbf{X}(t) = 0!$) is given in the Supporting Note S5, Eq.(32); it is a probability density function (where we changed variables to have a homogeneous equation, $\mathbf{Y} \rightarrow \mathbf{y} = \mathbf{Y} - (\mathbb{I} - \mathbf{A})^{-1} \mathbf{D}$),

$$\mathbf{f}_0(\mathbf{y}) = \frac{1}{\sqrt{(2\pi)^N |\sigma|}} \exp\left(-\frac{1}{2}(\mathbf{y}\sigma^{-1})\mathbf{y}\right) \quad . \quad (2)$$

The LRT prediction for the perturbed state is an expectation value computed using this probability density, which is a multivariate normal distribution centred on the solution of the unperturbed Leontief IO model, $(\mathbb{I} - \mathbf{A})^{-1} \mathbf{D}$, when expressed in terms of the variable \mathbf{Y} . For a step demand shock we get Eq. (9) as stationary solution ($t \rightarrow \infty$),

$$\langle \Delta Y_k \rangle_X = \rho_{ki} X_i, \quad \text{with } \rho_{ki} = \int_0^\infty (\sigma^{-1})_{ij} \langle Y_k(\tau) Y_j(0) \rangle_0 d\tau \quad , \quad (3)$$

where $\langle \cdot \rangle_0$ means that the expectation value is computed assuming \mathbf{f}_0 as underlying probability density function. We emphasize again that Eq. (3) shows how a *non-equilibrium* expectation value ($\langle \Delta Y_k \rangle_X$ for $\mathbf{X} \neq \mathbf{0}$) can be defined in terms of a known *equilibrium* expectation value, namely ρ_{ki} in Eq. (3).

The computation of output changes in a perturbed Leontief IO model would proceed along a different route. Let us again consider a step demand shock $\mathbf{X}(t) = \mathbf{X}\theta(t)$. We are now interested in the *equilibrium* state of an economy under such a demand shock. Therefore, we introduce the perturbed demand, \mathbf{D}_P as $\mathbf{D}_P = \mathbf{D} + \mathbf{X}$ and ask for the stationary solution of the stochastic differential equation,

$$\dot{\mathbf{Y}} = (\mathbf{A} - \mathbb{I})\mathbf{Y} + \mathbf{D}_P + \mathbf{F}(t) \quad . \quad (4)$$

Formally, Eqs. (1) and (4) are identical. However, the perturbed equilibrium perspective assumes a different stationary solution, namely

$$\mathbf{f}_P(\mathbf{z}) = \frac{1}{\sqrt{(2\pi)^N |\sigma_P|}} \exp\left(-\frac{1}{2}(\mathbf{z}\sigma_P^{-1})\mathbf{z}\right) \quad , \quad (5)$$

where $\mathbf{z} = \mathbf{Y} - (\mathbb{I} - \mathbf{A})^{-1} \mathbf{D}_P \neq \mathbf{Y} - (\mathbb{I} - \mathbf{A})^{-1} \mathbf{D}$ and $(\sigma_P)_{ij}(t) = \langle (z_i(t) - \langle z_i \rangle)(z_j(t) - \langle z_j \rangle) \rangle$ with $\sigma_P(t \rightarrow \infty) \equiv \sigma_P$. As could be expected, the stationary state of Eq.(4), i.e. the distribution of values of $\Delta \mathbf{Y}$ for $t \rightarrow \infty$, is now a multivariate normal distribution around the perturbed equilibrium state $(\mathbb{I} - \mathbf{A})^{-1} \mathbf{D}_P$. It is clear that this solution coincides with the LRT solution for $\mathbf{D}_P = \mathbf{D}$, i.e. in the absence of a shock, $\mathbf{X} = 0$. Also, only for $\mathbf{X} = 0$ would the correct expectation value to compute the susceptibility matrix and output change in LRT be given by the distribution $\mathbf{f}_P(\mathbf{z})$.

Within this note we have now encountered three different types of expectation value, namely (i) the equilibrium expectation value using measure \mathbf{f}_0 , $\langle \cdot \rangle_0$, (ii) the perturbed equilibrium expectation value given by \mathbf{f}_P , call it $\langle \cdot \rangle_P$, and (iii) the nonequilibrium expectation value $\langle \cdot \rangle_X$. These three expectation values coincide only in the absence of shocks, $\mathbf{X} = 0$. From a physical point of view, they describe three different types of system, namely a system relaxing to the state (i) \mathbf{f}_0 , (ii) \mathbf{f}_P , or (iii) \mathbf{f}_0 while responding to an external driving force, $\mathbf{X}(t)$ (a physicist would say that the external force \mathbf{X} “does work on the system”). In physics, the latter class of systems are closely related to “dissipative structures”, i.e. systems in a steady non-equilibrium state driven by an exchange of energy and/or matter with the environment.

In brief, while both the LRT and perturbed IO approach start from the same stochastic differential equation, Eqs. (1) and (4), they fundamentally differ in their definitions of expectation values. In contrast to expected values in the Leontief IO model in the perturbed equilibrium approach, the LRT approach assumes that the stationary solution of the system does *not* change after the application of an external shock.

REVIEWERS' COMMENTS:

Reviewer #2 (Remarks to the Author):

I would like to thank the authors for clarifying my doubts. I think that the new version of the manuscript is very clear, and I recommend it for publication in its current form.

REVIEWERS' COMMENTS:

Reviewer #2 (Remarks to the Author):

I would like to thank the authors for clarifying my doubts. I think that the new version of the manuscript is very clear, and I recommend it for publication in its current form.

OUR RESPONSE:

We also thank the referee for helping us to clarify this issue in our manuscript and are glad to see that it is now in an acceptable form.